# Metal-centred azaphosphatriptycene gear with a photo- and thermally driven mechanical switching function based on coordination isomerism

Hitoshi Ube[1], Yoshihiro Yasuda[1], Hiroyasu Sato[2] & Mitsuhiko Shionoya[1]

Metal ions can serve as a centre of molecular motions due to their coordination geometry, reversible bonding nature and external stimuli responsiveness. Such essential features of metal ions have been utilized for metal-mediated molecular machines with the ability to motion switch via metallation/demetallation or coordination number variation at the metal centre; however, motion switching based on the change in coordination geometry remain largely unexplored. Herein, we report a $Pt^{II}$-centred molecular gear that demonstrates control of rotor engagement and disengagement based on photo- and thermally driven cis–trans isomerization at the $Pt^{II}$ centre. This molecular rotary motion transmitter has been constructed from two coordinating azaphosphatriptycene rotators and one $Pt^{II}$ ion as a stator. Isomerization between an engaged cis-form and a disengaged trans-form is reversibly driven by ultraviolet irradiation and heating. Such a photo- and thermally triggered motional interconversion between engaged/disengaged states on a metal ion would provide a selector switch for more complex interlocking systems.

[1] Department of Chemistry, Graduate School of Science, The University of Tokyo, 7-3-1 Hongo, Bunkyo-ku, Tokyo 113-0033, Japan. [2] Rigaku Corporation, 3-9-12 Matubara-cho, Akishima, Tokyo 196-8666, Japan. Correspondence and requests for materials should be addressed to M.S. (email: shionoya@chem.s.u-tokyo.ac.jp).

Transmission of rotary motion is a key process of molecular machines[1–3]. To correlate two or more movable elements in a controllable manner, a stator, which can bring them appropriately close to each other, is a key part of motion. A significant number of excellent examples have been reported on synthetic molecular gearing systems with intramolecularly correlated rotators[4–9]. However, the control of rotary transmission between molecular rotators is still in an early phase[10,11], and therefore, in particular, *switchable* motion transmission is a challenge. Dynamic engagement between rotators is a typical rotary transmission in molecular machines. Triptycene is a well-known part of gear molecules with a rigid, highly symmetrical paddlewheel structure. More than one triptycene rotators can be covalently connected with an organic stator designed so as to have a proper positional relationship between the connected rotators[12–21]. Another unique example of covalently linked systems is a silicon-centred bistriptycene system[19], which undergoes switchable gearing triggered by a chemical stimulus, fluorination/defluorination. We focused on metal ions as a control element of molecular motions due to their essential features such as reversible bonding natures with ligands, dynamic ligand exchange and external stimuli responsiveness[22–24]. A certain number of excellent examples of metal-mediated molecular machines capable of motion switching via metallation/demetallation or coordination number variation on the metal centre have been reported[25–27].

Herein, we report a $Pt^{II}$-centred molecular gear that demonstrates control of rotor engagement and disengagement based on photo- and thermally driven *cis–trans* isomerization at the $Pt^{II}$ centre. This molecular rotary motion transmitter has been constructed from two coordinating azaphosphatriptycene rotators and one $Pt^{II}$ ion as a stator. Isomerization between an engaged *cis*-form and a disengaged *trans*-form is reversibly driven by ultraviolet irradiation and heating. Such a photo- and thermally triggered motional interconversion between engaged/disengaged states on a metal ion would provide a selector switch for more complex interlocking systems.

## Results

**Design of metal-centred molecular gear.** In this study, we have developed a metal-centred molecular gear $PtCl_2\mathbf{1}_2$, in which two ligands as rotators, 2-methoxy-9-aza-10-phosphatriptycene (**1**), directly bind to the $Pt^{II}$ stator[28,29]. One striking feature of this system is a clutch-like function that allows switching of the engagement of the two rotators based on photo- and thermally driven *cis–trans* isomerization on the $Pt^{II}$ centre in a traceless manner with no chemical by-products (Fig. 1)[30,31].

**Synthesis of azaphosphatriptycene ligand.** 2-Methoxy-9-aza-10-phosphatriptycene (**1**), in which the bridgehead positions of triptycene are replaced by a nitrogen and a phosphorus atoms, was chosen as a ligand-type rotator[32]. The rotator **1** was synthesized from 3-anisidine in four steps in 9% overall yield, and was characterized by $^{1}H$, $^{13}C$ and $^{31}P$ nuclear magnetic resonance (NMR) spectroscopy, electrospray ionization-mass spectrometry and elemental analysis (Supplementary Figs 1–10). We expected that when two rotators **1** as monodentate phosphine ligands bind to a compatible $Pt^{II}$ ion in a *cis* position, they should gear with each other.

**Preparation and characterization of $Pt^{II}$ complexes.** The reaction of rotator **1** and 0.5 eq of $K_2PtCl_4$ in $EtOH/H_2O$ (1/1, v v$^{-1}$) at room temperature for 21 h afforded a mixture of square-planar $Pt^{II}$–phosphine complexes, *cis*-$PtCl_2\mathbf{1}_2$ and a small amount of *trans*-$PtCl_2\mathbf{1}_2$ (Figs 1,2a). After recrystallization from $CHCl_3/$

diethyl ether, pure *cis*-$PtCl_2\mathbf{1}_2$ was successfully obtained as colourless crystals in 59% yield (Supplementary Figs 11 and 13). The coordinating donor atom and the geometry around the $Pt^{II}$ centre in solution were determined by NMR spectroscopy. In a $^{31}P$ NMR spectrum of the *cis* isomer in $C_6D_6$, a $^{31}P$-$^{195}Pt$ coupling was observed ($J_1 = 3{,}625$ Hz; Supplementary Fig. 12) due to the binding of phosphine ligand to the central $Pt^{II}$ ion. Then, a $^{1}H$ NMR spectrum of the *cis* isomer showed that the proton signals for the 3-positions of azaphosphatriptycene shifted upfield from the free ligand, indicating that two azaphosphatriptycene ligands were close to each other. Moreover, a lineshape analysis of the spectral pattern of the proton signals of the 4-positions suggests that there are two rotational isomers, *meso* and *dl* forms, in a 1:2 ratio in solution (Fig. 2b, Supplementary Fig. 14). We analysed the kinetics of the gear slippage during interconversion between *meso* and *dl* isomers of *cis*-$PtCl_2\mathbf{1}_2$. Variable temperature $^{1}H$ NMR measurement of *cis*-$PtCl_2\mathbf{1}_2$ was described by the two-signal overlap model of the *meso* and *dl* isomers. An Eyring plot of the exchange rates of the isomers at every temperature gave activation parameters of the interconversion, $\Delta H^{\ddagger} = 16.4 \pm 0.2$ kcal·mol$^{-1}$ and $\Delta S^{\ddagger} = -0.9 \pm 0.7$ cal·K$^{-1}$ (Fig. 2c, Supplementary Fig. 15). The enthalpy value is significantly lower than the reported value for covalently connected triptycene gears[15,19].

It is noteworthy in connection with motion transmission that a certain number of $Pt^{II}$ complexes display photo-driven *cis/trans* isomerization[30,31]. Under photoirradiation at 360 nm, *cis*-$PtCl_2\mathbf{1}_2$ was found to be isomerized to *trans*-$PtCl_2\mathbf{1}_2$ in $C_6H_6$ at room temperature (*cis/trans* = 50:50 by $^{1}H$ NMR in $CDCl_3$ at 300 K). As a result of slow evaporation, *trans*-$PtCl_2\mathbf{1}_2$ was isolated as yellow crystals in 29% yield (Supplementary Figs 16 and 17). In a $^{31}P$ NMR spectrum of the *trans* isomer in $CDCl_3$, a $^{31}P$-$^{195}Pt$ coupling ($J_1 = 2{,}937$ Hz, in Supplementary Fig. 18) indicated that the two phosphine ligands bind to the central $Pt^{II}$ ion. On the other hand, its $^{1}H$ NMR spectrum in $CDCl_3$ showed neither splitting nor upfield shift of the proton signals for the 4-positions, suggesting the absence of significant intramolecular interactions between the two rotators.

**X-ray crystallographic analyses.** Crystals of *cis*-$PtCl_2\mathbf{1}_2$·(ether) suitable for single-crystal X-ray structure analysis were obtained by liquid–liquid diffusion of $Et_2O$ into a solution of $PtCl_2\mathbf{1}_2$ in toluene. One molecule of diethyl ether was included into the unit structure (Fig. 3, Supplementary Fig. 19). The X-ray diffraction data demonstrated that the two rotators adopt an 'engaged' *cis* form in the solid state. A unit cell consists of a *meso* isomer of $PtCl_2\mathbf{1}_2$, one rotational isomer comes from tight meshing of the two rotators. Notably, intramolecular CH–π interactions were observed between the two rotators. Each $Pt^{II}$ ion is in a distorted square planar geometry, in which the P–Pt–P angle is over 90° (99.95(5)°) due to the bulkiness of rotator **1**. The two rotators **1** are thus suitably engaged with each other on the $Pt^{II}$ stator. In contrast, single-crystal X-ray structure analysis of *trans*-$PtCl_2\mathbf{1}_2$·$(C_6H_6)_2$ revealed that the two phosphine ligands as rotators are across from one another in the square planar $Pt^{II}$ complex (Supplementary Fig. 20). This photoisomerized *trans* form can be regarded as a 'disengaged' state of the metal-centred molecular gear.

**Photo- and thermally driven isomerization of $PtCl_2\mathbf{1}_2$.** We then envisioned that this gear system could be applied to a stimuli-responsive molecular switch based on the photo- and thermally driven *cis–trans* isomerization in an appropriate solvent. It is well known that photo or thermal isomerization of diphosphine

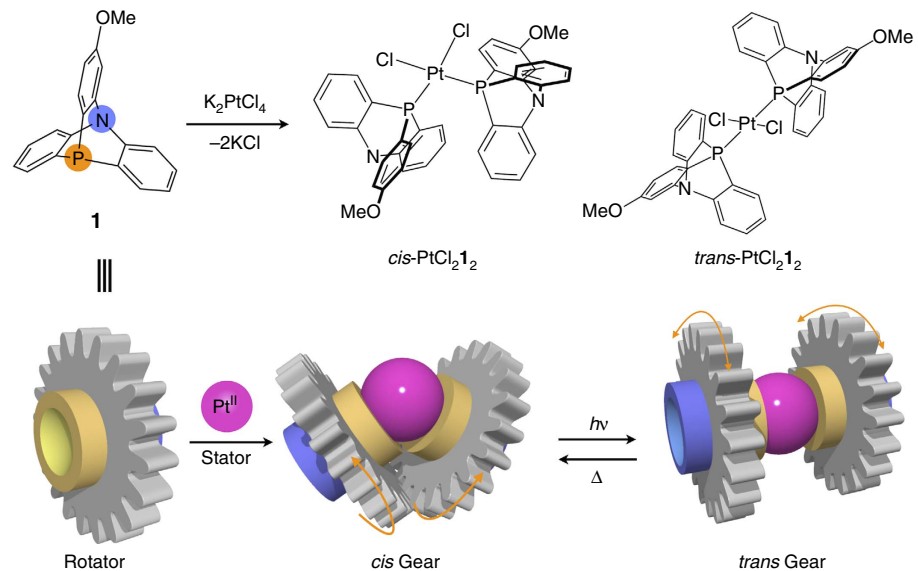

**Figure 1 | Schematic representation of a Pt$^{II}$-centred molecular gear PtCl$_2$1$_2$.** This molecular gear has two azaphosphatriptycene rotators coordinating to the central Pt$^{II}$ ion as a stator. Isomerization between an engaged *cis*-form and a disengaged *trans*-form are reversibly driven by ultraviolet irradiation at 360 nm and heating.

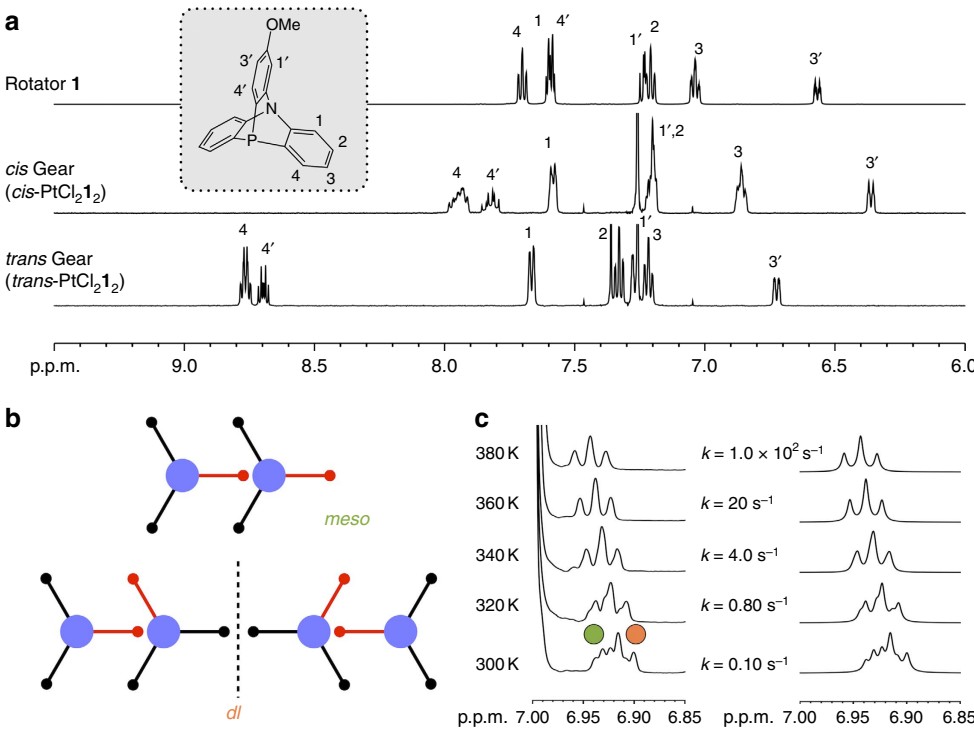

**Figure 2 | $^1$H NMR spectra of rotator 1 and *cis*- and *trans*-PtCl$_2$1$_2$.** The signals of a methoxy group of rotator **1** (∼3.8 p.p.m.) are omitted for clarity. For the whole NMR spectra, see the Supplementary Figures 7, 11 and 16. (**a**) $^1$H NMR spectrum of (i) **1**, (ii) *cis*-PtCl$_2$1$_2$ and (iii) *trans*-PtCl$_2$1$_2$ (500 MHz, CDCl$_3$, 300 K). The $^1$H NMR spectra of *cis*-PtCl$_2$1$_2$ include *meso* and *dl* isomers in a ∼1:2 ratio. (**b**) Isomerism based on rotational conformation. (**c**) Observed and simulated spectra of 3-positions' proton at varied temperatures (500 MHz, TCE-$d_2$/toluene-$d_8$ = 1:1). Left: observed spectra in the range from 380 to 300 K. Right: simulated spectra based on the two-state exchange model. Green and orange circles denote *meso* and *dl* isomers, respectively.

Pt$^{II}$ complexes highly depends on the solvent polarity[8]. Polar solvents generally prefer *cis* form rather than *trans* form because the *cis* complex has a dipole moment that interacts better with the solvent polarity. Photo-driven isomerization from *cis* to *trans* form was then examined in a solvent with low polarity. Ultraviolet light at 360 nm was irradiated to a solution of pure *cis*-PtCl$_2$1$_2$ in toluene-$d_8$ at room temperature (Fig. 4a,b). A photo stationary state was reached after 30 min, where the *cis*/*trans* ratio was changed to 15:85 (Supplementary Fig. 21). On the other hand, in more polar 1,1,2,2,-tetrachloroethane-$d_2$ (TCE-$d_2$), thermal isomerization from *trans* to *cis* was so fast in the dark at room

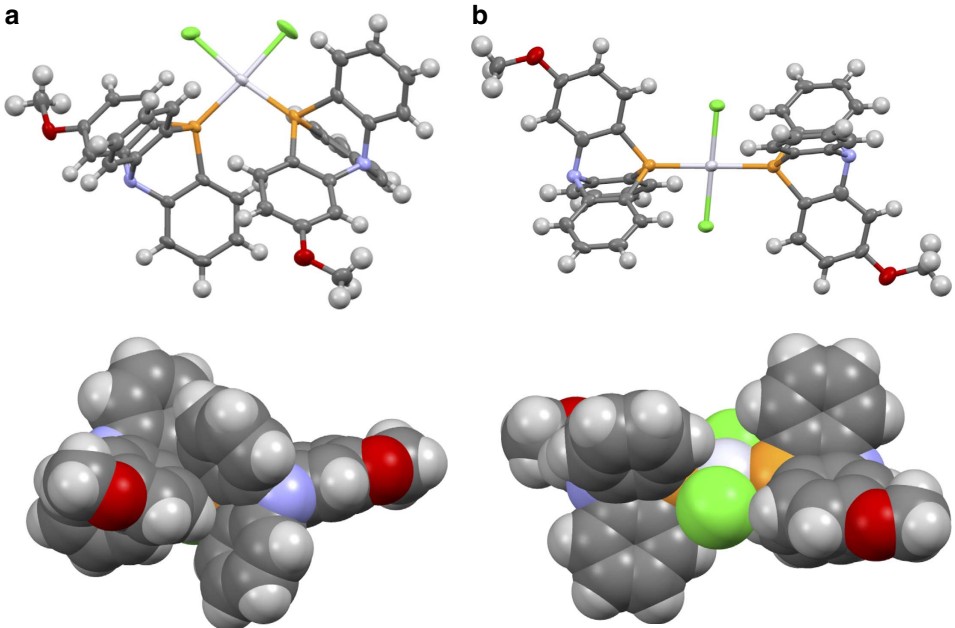

**Figure 3 | X-ray crystal structures of *cis*- and *trans*-PtCl₂1₂.** (**a**) *cis*-PtCl₂**1**₂ · (ether). (**b**) *trans*-PtCl₂**1**₂ · (C₆H₆)₂. In both cases, the structures are indicated as ORTEP (Oak Ridge Thermal Ellipsoid Plot) diagram with 50% thermal ellipsoid (upper) and space-filling model (bottom). Solvents are omitted for clarity, and colours are coded according to CPK (Corey, Pauling, Koltun) colouring.

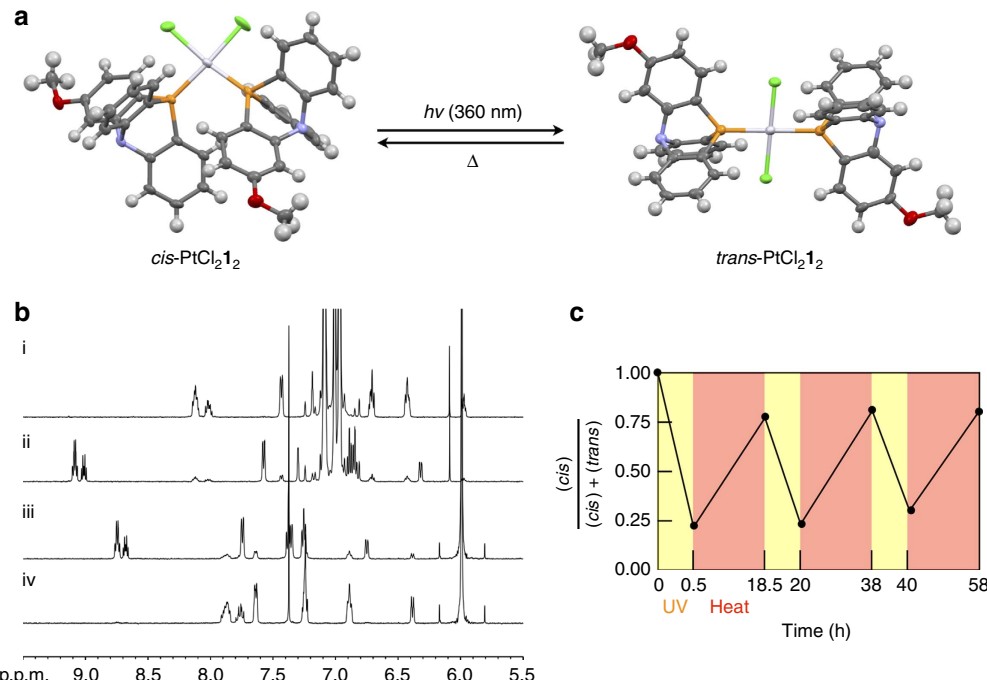

**Figure 4 | Photo and thermal switching of molecular gear PtCl₂1₂.** (**a**) Photo- and thermally induced isomerization between *cis*- and *trans*-PtCl₂**1**₂. (**b**) ¹H NMR spectra for photo- and thermally induced isomerization from *cis*- to *trans*-PtCl₂**1**₂ (500 MHz, 300 K). (i) A solution of single crystals of *cis*-PtCl₂**1**₂ in toluene-*d₈* (*cis/trans* = 99:1); (ii) a solution of (i) after photoirradiation at 360 nm (after 30 min, *cis/trans* = 15:85); (iii) a solution of single crystals of *trans*-PtCl₂**1**₂ in TCE-*d₂* (after 5 min, *cis/trans* = 12:88); (iv) a solution of (iii) after 10 h at room temperature (*cis/trans* = 2:98). (**c**) Reversible switching of the molecular gearing system, PtCl₂**1**₂, in TCE-*d₂*/toluene-*d₈* = 1:1 (v v⁻¹).

temperature that it was difficult to obtain a ¹H NMR spectrum of pure *trans* complex in the TCE-*d₂* solution because of the rapid isomerization to the *cis* form. After 10 h, the *trans* complex was transformed into *cis* form nearly quantitatively (*cis/trans* = 98:2; Supplementary Fig. 22). This structural interconversion was repeatable in a mixed solvent of TCE-*d₂*/toluene-*d₈* = 1:1. When a solution of pure *cis*-PtCl₂**1**₂ was irradiated by ultraviolet light at 360 nm, the *cis* to *trans* conversion proceeded smoothly at room temperature, and the reaction achieved its equilibrium (*cis/ trans* = 19:81) in 30 min. In this mixed solvent system, the interconversion from *trans* to *cis* was slow enough to determine the ratio of the complex by NMR at 300 K. When the *trans*-based

solution was heated at 100 °C for 10 h, the *trans*-based solution was reversed to the *cis*-based solution with the *cis/trans* = 78:22. This *cis–trans* isomerization process was thus repeatable at least three times by the repetition of stimuli (Fig. 4c and Supplementary Figs 23 and 24).

## Discussion

In conclusion, we have developed a molecular gear, PtCl$_2$**1**$_2$, composed of two azaphosphatriptycene rotators **1** with a Pt$^{II}$ ion acting as a stator. The repeatable mechanical switching function based on the *cis–trans* isomerization at the Pt$^{II}$ centre was achieved by photoirradiation and heating. Traceless external stimuli-responsive configurational changes of metal ions show promise as a movement element of molecular machines with a motion transmission function.

## Methods

**General information.** Unless otherwise noted, solvents and reagents were purchased and used without further purification. 2-Methoxy-9-aza-10-phospha-triptycene (**1**) was synthesized from 3-anisidine (Supplementary Figs 1–10 and Supplementary Note 1).

$^1$H, $^{13}$C, $^{31}$P NMR and other two-dimensional NMR spectra were recorded on a Bruker AVANCE III-500 (500 MHz) spectrometer. Tetramethylsilane was used as an internal standard ($\delta$ 0 p.p.m.) for $^1$H and $^{13}$C NMR measurements when CDCl$_3$ was used as solvent. A residual solvent signal was used for calibration of $^1$H NMR measurements when other deuterated solvents (C$_6$HD$_5$: 7.16 p.p.m.; toluene-$d_7$: 6.97 p.p.m.; 1,1,2,2-tetrachloroethane-$d$: 5.99 p.p.m.) was used as a solvent. ESI-TOF mass data were recorded on a Micromass LCT Premier XE mass spectrometer. Unless otherwise noted, experimental conditions are as follows: ion mode, positive; capillary voltage, 3,000 V; sample cone voltage, 30 V; desolvation temperature, 150 °C; source temperature, 80 °C. Melting point was measured by Yanaco Micro Melting Point Apparatus MP-500D and uncorrected. Elemental analysis was conducted in the Microanalytical Laboratory, Department Chemistry, Graduate School of Science, the University of Tokyo (Tokyo, Japan). Infrared spectra were recorded on a Jasco FT/IR 4,200 with an ATR equipment.

**Synthesis of *cis*-PtCl$_2$1$_2$.** To a 5.0 mM solution of K$_2$PtCl$_4$/H$_2$O (40 ml, 0.20 mmol, 1.0 eq) was added a 10 mM solution of 2-methoxy-9-aza-10-phospha-triptycene (**1**) in EtOH (40 ml, 0.40 mmol, 2.0 eq). The suspended solution was then stirred at room temperature for 21 h in the dark. The resulting precipitate was collected by filtration, washed with H$_2$O and EtOH, and dried under vacuum to give a colourless solid (144 mg). The crude product was purified by recrystallization from chloroform/diethyl ether to give *cis*-PtCl$_2$**1**$_2$ (112 mg, 0.118 mmol, 59%) as a colourless solid. $^1$H NMR (C$_6$D$_6$, 500 MHz, 300 K): $\delta$ 7.81–7.78 (m, 4H), 7.71–7.66 (m, 2H), 7.05 (d, $J$ = 7.6 Hz, 4H), 6.83 (d, $J$ = 7.6 Hz, 4H), 6.31–6.27 (m, 4H), 6.04–6.00 (m, 4H), 5.62–5.68 (m, 2H), 2.57 (s, 2H), 2.55 (s, 4H); $^{31}$P NMR (C$_6$D$_6$, 202 MHz, 300 K): $\delta$ − 31.0 ($J_{P–Pt}$ = 3,625 Hz); HRMS (CHCl$_3$/CH$_3$CN/HCO$_2$H, positive): [PtCl**1**$_2$]$^+$ (C$_{38}$H$_{28}$ClN$_2$O$_2$P$_2$Pt) $m/z$ 836.0958 (required, 836.0957).

**Synthesis of *trans*-PtCl$_2$1$_2$.** In a 50 ml three-necked flask, a solution of *cis*-PtCl$_2$**1**$_2$ (20.0 mg, 21 µmol) in benzene was irradiated at 360 nm for 1 h at room temperature. The solvent was removed by evaporation to give a yellow solid (22.5 mg). The crude product was recrystallized from benzene (2 ml) to obtain yellow crystals of *trans*-PtCl$_2$**1**$_2$ · (C$_6$H$_6$)$_2$. After dryness, 5.8 mg of desired complex was obtained, which contains 0.5 eq of benzene (confirmed by NMR, 6.1 µmol, 29%). $^1$H NMR (CDCl$_3$, 500 MHz, 300 K): $\delta$ 8.78–8.75 (m, 4H), 8.70 (dt, $J$ = 8.4, 5.4 Hz, 2H), 7.66 (dd, $J$ = 7.7, 1.1 Hz, 4H), 7.34 (td, $J$ = 7.6, 1.2 Hz, 4H), 7.28–7.25 (m, 2H), 7.22 (td, $J$ = 7.5, 1.3 Hz, 4H), 6.72 (dt, $J$ = 8.4, 1.2 Hz, 2H), 3.82 (s, 6H); $^{31}$P NMR (CDCl$_3$, 202 MHz, 300 K): $\delta$ − 39.0 ($J_{P–Pt}$ = 2,937 Hz); HRMS (CHCl$_3$/CH$_3$CN/HCO$_2$H, positive): [PtCl**1**$_2$]$^+$ (C$_{38}$H$_{28}$ClN$_2$O$_2$P$_2$Pt) $m/z$ 836.0958 (required, 836.0957).

**Photo- and thermally driven isomerization of PtCl$_2$1$_2$.** A 1.0 mM solution of *cis*-PtCl$_2$**1**$_2$ in TCE-$d_2$/toluene-$d_8$ = 1:1 (600 µl, 0.60 µmol) and a 100 mM solution of 1,4-dioxane in TCE-$d_2$/toluene-$d_8$ = 1:1 (6.0 µl, 0.60 µmol) were placed in an NMR tube, which was sealed by a septum rubber and degassed by freeze–pump–thaw three times. The reaction mixture was irradiated with ultraviolet lamp (ASAHI, MAX-303) using 360 nm filter (bandwidth = 10 nm) at room temperature, and was heated at 100 °C in the dark (Supplementary Figs 19–22).

**X-ray diffraction analysis.** Single-crystal X-ray crystallographic analyses were performed using a Rigaku Saturn724+ diffractometer with MoKα radiation (for *cis*-PtCl$_2$**1**$_2$) or Rigaku RAXIS-RAPID imaging plate diffractometer with MoKα radiation (for *trans*-PtCl$_2$**1**$_2$), and obtained data were calculated using the Crystal Structure crystallographic software package except for refinement,

which was performed using SHELXL-2014 (ref. 33). All hydrogen atoms were placed geometrically and refined using a riding model. Details for the synthesis and X-ray diffraction mesurements of both *cis*- and *trans*-complex are given in CIF files and Supplementary Figs 19 and 20 and Supplementary Note 2.

**Data availability.** Crystallographic data in this paper can be obtained free of charge from the Cambridge Crystallographic Data Centre (http://www.ccdc.ca-m.ac.uk/data_request/cif). The Deposit numbers are 1404948 (*cis*-PtCl$_2$**1**$_2$) and 1404949 (*trans*-PtCl$_2$**1**$_2$), respectively. All other data are available on this article and its Supplementary Information file.

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

## Acknowledgements

This study was supported by JSPS KAKENHI Grant Numbers JP26248016 and JP16H06509 (Coordination Asymmetry).

## Author contributions

H.U., Y.Y. and M.S. conceived and designed the experiments and analysed the data. Y.Y. performed the experiments. H.U. and H.S. measured and solved the X-ray crystallographic analyses. H.U. and M.S. prepared the manuscript.

## Additional information

**Competing financial interests:** The authors declare no competing financial interests.

