## [Peer review file · Nature Communications]

REVIEWERS' COMMENTS:

Reviewer #1 (Remarks to the Author):

I am glad to see that the crystallographic problems for this paper have been largely corrected and from this perspective I can recommend publication.

Nonetheless, I remain quite adamant about the portion of my original review that found that the casting of this paper in mechanical guises strongly detracted from the seriousness which surely the authors expect. It all strikes me as somewhat juvenile and unnecessary. However, if they are willing to endure this response from readers, I am willing to let it go.

A List of Response to the Referee's Comments

Reviewer #1 (Remarks to the Author):

I am glad to see that the crystallographic problems for this paper have been largely corrected and from this perspective I can recommend publication.

Nonetheless, I remain quite adamant about the portion of my original review that found that the casting of this paper in mechanical guises strongly detracted from the seriousness which surely the authors expect. It all strikes me as somewhat juvenile and unnecessary. However, if they are willing to endure this response from readers, I am willing to let it go.

Thank you very much for your kind comments and recommendation for publication in *Nature Communications*. According to your suggestion, we substituted the term “clutch” for “mechanical switching” in the title. For other related changes we made, please see the parts highlighted in yellow.